# Collagen methionine sulfoxide and glucuronidine/LW-1 are markers of coronary artery disease in long-term survivors with type 1 diabetes. The Dialong study

**Kristine B. Holte** [1,2]*, **Mona Svanteson**[2,3], **Kristian F. Hanssen**[2,4], **Kari Anne Sveen**[1,2], **Ingebjørg Seljeflot**[2,5], **Svein Solheim**[5], **David R. Sell**[6], **Vincent M. Monnier**[6], **Tore Julsrud Berg**[1,2]

1 Department of Endocrinology, Morbid Obesity and Preventive Medicine, Oslo University Hospital, Oslo, Norway, 2 Institute of Clinical Medicine, Faculty of Medicine, University of Oslo, Oslo, Norway, 3 Department of Radiology and Nuclear Medicine, Oslo University Hospital, Oslo, Norway, 4 The Norwegian Diabetics' Center, Oslo, Norway, 5 Center for Clinical Heart Research, Department of Cardiology, Oslo University Hospital, Oslo, Norway, 6 Department of Pathology and Biochemistry, Case Western Reserve University School of Medicine, Cleveland, Ohio, United States of America

* kribechholte@gmail.com

**Data Availability Statement:** Data cannot be shared publicly because of ethical and legal reasons. The Norwegian Regional Committees for

## Abstract

### Objectives

Type 1 diabetes is a risk factor for coronary heart disease. The underlying mechanism behind the accelerated atherosclerosis formation is not fully understood but may be related to the formation of oxidation products and advanced glycation end-products (AGEs). We aimed to examine the associations between the collagen oxidation product methionine sulfoxide; the collagen AGEs methylglyoxal hydroimidazolone (MG-H1), glucosepane, pentosidine, glucuronidine/LW-1; and serum receptors for AGE (RAGE) with measures of coronary artery disease in patients with long-term type 1 diabetes.

### Methods

In this cross-sectional study, 99 participants with type 1 diabetes of $\geq$ 45-year duration and 63 controls without diabetes had either established coronary heart disease (CHD) or underwent Computed Tomography Coronary Angiography (CTCA) measuring total, calcified and soft/mixed plaque volume. Skin collagen methionine sulfoxide and AGEs were measured by liquid chromatography-mass spectrometry and serum sRAGE/esRAGE by ELISA.

### Results

In the diabetes group, low levels of methionine sulfoxide (adjusted for age, sex and mean HbA$_{1c}$) were associated with normal coronary arteries, OR 0.48 (95% CI 0.27–0.88). Glucuronidine/LW-1 was associated with established CHD, OR 2.0 (1.16–3.49). MG-H1 and glucuronidine/LW-1 correlated with calcified plaque volume (r = 0.23–0.28, p<0.05), while

Medical and Health Research Ethics - South East (project no. 2014/851) and local personal data protection rules only allow for the data to be stored in prespesified research server, but in certain cases data can be transferred to a person if the data will appear anonymized for the recipient. Although the data are de-identified, due to the relative small number of patients with type 1 diabetes (100) from one single centre, the data could potentially be linked back to a person. An ethically compliant dataset may be provided by the authors by contacting Dr. Elisabeth Qvigstad, Head of Research at the Department of Endocrinology, Oslo University Hospital (eqwigsta@ous-hf.no / elisabeth.qvigstad@medisin.uio.no).

**Funding:** This work was supported by the Oslo Diabetes Research Centre, the Norwegian Diabetics' Centre, The Family Blix Foundation, and The Odd Fellow Foundation. The funders had no role in study design, data collection and analysis, decision to publish, or preparation of the manuscript.

**Competing interests:** The authors have declared that no competing interests exist.

pentosidine correlated with soft/mixed plaque volume (r = 0.29, p = 0.008), also in the adjusted analysis.

## Conclusions

Low levels of collagen-bound methionine sulfoxide were associated with normal coronary arteries while glucuronidine/LW-1 was positively associated with established CHD in long-term type 1 diabetes, suggesting a role for metabolic and oxidative stress in the formation of atherosclerosis in diabetes.

## 1. Introduction

Patients with long-term type 1 diabetes have a high prevalence of coronary artery disease (CAD). The accelerated coronary atherosclerosis seen in diabetes is more severe and diffuse than in people without diabetes [1–3]. However, some people have normal coronary arteries after 50 years of type 1 diabetes as we have recently reported [4].

Why some people with long-term type 1 diabetes remain free of CAD is not known, but one theory by which hyperglycemia may lead to CAD is partly through the formation of oxidation products and advanced glycation end-products (AGEs) resulting in structural and functional alterations in the arterial wall [1]. High mean $HbA_{1c}$ level is a risk factor for future cardiovascular events [5, 6], and rodent studies suggest that reactive oxygen species (ROS) and carbonyl intermediates in AGEs formation play an essential role in linking hyperglycemia to cardiovascular disease [7, 8].

ROS are partially reduced unstable oxygen metabolites that easily react with other molecules in a cell. The oxidation product methionine sulfoxide (MetSO) can be formed through ROS-mediated protein methionine oxidation. This reaction may cause important structural and functional disruptions to the target protein, and it has been suggested that protein methionine oxidation may have a pathogenic role in vascular disease [9]. The collagen-bound MetSO has previously been linked to vascular complications in type 1 diabetes [10, 11].

AGEs are a heterogeneous group of compounds formed through several steps including the non-enzymatic reaction between a reducing sugar/reactive dicarbonyl and an amino group. Collagen-bound methylglyoxal hydroimidazolone 1 (MG-H1), pentosidine and glucuronidine/LW-1 are all examples of AGEs that have been linked with future subclinical macrovascular disease [12].

Previous studies in type 1 diabetes examining the associations between collagen oxidation products and AGEs with cardiovascular disease are based on either the presence of previous cardiovascular events or surrogate markers for CAD such as measurement of the carotid intima media thickness or coronary artery calcification score (CAC). Computer Tomography Coronary Angiography (CTCA) is a non-invasive technique that has the advantage of being able to differentiate between absence of CAD and gradually worsening degree of coronary atherosclerosis. The plaque morphology identified also provides valuable information [13]. The associations between MetSO and AGEs with CAD and plaque morphology as defined by CTCA in type 1 diabetes have not been reported previously.

We therefore aimed to explore associations between the collagen-bound oxidation product MetSO, four specific collagen-bound AGEs and serum receptors for AGE (RAGEs) with normal, plaque-free, coronary arteries on CTCA, and with measures of coronary artery disease

including plaque morphology on CTCA and established coronary heart disease (CHD) in patients with long-term type 1 diabetes compared to controls.

## 2. Subjects, materials and methods

### 2.1 Study design and subjects

The Dialong study was a cross-sectional study of long-term survivors of type 1 diabetes conducted in 2015, including patients with type 1 diabetes for ≥ 45 years and an appropriate control group of similar age and sex distribution. The inclusion criteria have previously been set out in full [14]. Briefly, we invited all patients with type 1 diabetes diagnosed ≤ 1970 attending a state-funded specialized type 1 diabetes clinic in Oslo, Norway. Out of 136 eligible patients, 103 joined the CAD sub-study. The control group (n = 63) without diabetes consisted of spouses/friends of the patients with diabetes, and was included in the same time-period. First-degree relatives were excluded. The Norwegian Regional Committees for Medical and Health Research Ethics—South East (project no. 2014/851) approved the study, and all participants signed an informed consent.

### 2.2 Procedures

Background data were collected from patient charts at NDC, interviews and clinical examination [14]. All participants had a skin punch biopsy for measurements of collagen-linked AGEs and MetSO, fasting morning blood and urine samples, and retinal photos. The participants without established CHD (n = 88/103 in the diabetes group and n = 60/63 of controls) were referred to CTCA as previously described [4].

### 2.3 Outcomes

CTCA was performed on a Dual Source CT scanner (Somatom Definition Flash, Siemens, Erlangen, Germany), and the details for the investigation have been previously described [4].

*Normal coronary arteries* were defined as no plaque in any of the coronary arteries on CTCA. *Non-obstructive CAD* was defined as plaques resulting in a 1–50% diameter stenosis in any of the coronary arteries but none > 50%, and *obstructive CAD* as > 50% stenosis in any of the coronary arteries. *Established CHD* was defined as a previous episode of acute coronary syndrome or revascularization procedure, prior to enrolment in the study.

The volume (mm$^3$) of all plaques in each patient was calculated and differentiated in categories based on plaque morphology; *total plaque volume*, *calcified plaque volume* and *soft/mixed plaque volume*. The plaque morphology was determined in each plaque by the amount of calcifications (i.e. density > 130 Hounsfield units) present. A plaque containing more than 90% calcifications was defined as a calcified plaque and 0–90% as a soft/mixed plaque [15].

*CAC* was calculated according to the Agatston method [15, 16].

*Retinopathy* was defined as either background or proliferative retinopathy changes based on retina photos (wide-angle camera based on SLO-technique (Optos Daytona)), which were all analyzed by one certified ophthalmologist [14].

### 2.4 Skin collagen AGEs and MetSO and skin autofluorescence

A 3 mm skin punch biopsy was obtained from each participant from the upper lateral part of the nates. The sample was immediately placed in a sterile container, flushed with nitrogen gas and transferred to a temperature of -80 degrees Celsius. Skin collagen bound MetSO and AGEs (glucosepane, pentosidine, glucuronidine/fluorophore LW-1, and MG-H1) were measured using liquid chromatography tandem mass spectrometry (LC/MS/MS) as previously

described [14]. Briefly, samples of 50 ug insoluble delipidated skin collagen samples were exhaustively digested with proteolytic enzymes to release the free AGEs. The samples were spiked with isotopically labeled internal standards for each AGE and quantitated by LC/MS/MS. Each chromatography peak was individually analyzed in reference to the labeled peak. Collagen content was determined using the hydroxyproline assay as previously described [17]. Data are expressed in pmol AGE/mg collagen. The analyst was blinded regarding group affiliation. All participants also had skin intrinsic fluorescence measured non-invasively in the forearm by the autofluorescence reader (AGE reader) [18].

## 2.5 Serum sRAGE and esRAGE

Fasting blood samples without additives were separated within 1 hour by centrifugation at 2500xg for 10 min, and serum was transferred to a temperature of—80˚C until analysis. Serum sRAGE and esRAGE were quantified using enzyme-linked immunosorbent assay (ELISA) kits (Quantikine ELISA, R&D Systems, Abingdon, Oxon, UK and B-Bridge International Inc., Cupertino, CA, USA respectively) as directed by the manufacturer's instructions. Again, the analyst was blinded regarding group affiliation. The inter-assay coefficients of variation in our laboratory (at Oslo University Hospital, Ulleval) were 5.8% (sRAGE) and 5.7% (esRAGE).

## 2.6 Statistical analysis

The power analysis suggested we needed 77 participants with diabetes to detect a significant difference in glucosepane levels between patients with type 1 diabetes with obstructive CAD and patients with normal coronary arteries (power 90%, probability 0.05). This was based on a distribution in diabetic patients of 30%, 40% and 30% with absent CAD, non-obstructive CAD and obstructive CAD respectively and our own previous analysis showing mean glucosepane levels from the Oslo study to be 500 pmol/L higher in the group with obstructive CAD versus absent CAD [19, 20]. While our power analysis was based on only participants with obstructive disease and normal arteries, we decided to keep all the patients with non-obstructive CAD (60%) in our analyses to avoid loss of power. Cases with missing data were excluded, and we did a complete case analysis.

Clinical characteristics, CTCA findings and MetSO/AGEs/RAGEs levels were compared between the groups using two-tailed Student's t-test or Mann Whitney U test for continuous data as appropriate and $\chi^2$ for categorical data. For bivariate outcomes, logistic regression analyses were performed to adjust for possible confounders (age, sex, and mean HbA1c). Statin use was not included in the analyses as all participants with established CHD were expected to be on statins. Standardized values for MetSO and the AGEs were calculated. We performed Spearman correlation analyses to assess for correlations between the markers and the plaque volume measures, and linear regression analyses to adjust for confounders. We assessed for collinearity between variables and only included variables with r < 0.7 in the same model. Any dependent variable that did not have normally distributed residuals, were natural log (ln)-transformed. A p-value ≤ 0.05 was considered significant. All analyses were performed using IBM SPSS Statistics version 25 (IBM SPSS Inc., Armonk, NY: IBM Corp.).

## 3. Results

### 3.1 Clinical characteristics

Table 1 shows the clinical characteristics, the CTCA findings and the AGEs, MetSO and RAGE levels in the diabetes and control groups. Four of the participants in the diabetes group did not have a punch biopsy performed (due to needle phobia and previous allergic reaction to

**Table 1. Participant characteristics.**

| | Type 1 diabetes (n = 99) | Controls (n = 63) | P |
|---|---|---|---|
| *Demographics/risk factors* | | | |
| Age | 62.1 ± 7.1 | 62.7 ± 7.0 | 0.64 |
| Sex, male | 51 (51.5) | 28 (44.4) | 0.38 |
| Daily smoker | 4 (4) | 6 (9.5) | 0.16 |
| LDL-cholesterol, mmol/L | 2.7 (0.8) | 3.8 (1.0) | **< 0.001** |
| Blood pressure systolic | 146 ± 20 | 138 ± 20 | **0.012** |
| diastolic | 75 ± 8 | 81 ± 10 | **< 0.001** |
| eGFR[a] | 84 ± 19 | 82 ± 14 | 0.36 |
| CRP, mg/L, median (IQR) | 1.7 (0.8–3.5) | 1.2 (0.6–2.6) | **0.04** |
| *Diabetes related factors* | | | |
| EFD HbA$_{1c}$ %, mmol/mol | 8.0 ± 0.8 | | |
| | 63.5 ± 8.6 | | |
| Diabetes duration, years median (IQR) | 49 (47–54) | | |
| Persistent albuminuria | 17 (17.2) | | |
| Retinopathy None | 5 (5.1) | | |
| Background | 50 (50.5) | | |
| Proliferative | 44 (44.4) | | |
| Neuropathy | 63 (63.6) | 11 (17.5) | **< 0.001** |
| *Presence of CAD* | | | |
| Normal coronary arteries on CTCA | 14 (14.1) | 30 (47.6) | **< 0.001** |
| Non-obstructive CAD on CTCA | 50 (50.5) | 24 (38.1) | |
| Obstructive CAD on CTCA | 20 (20.2) | 6 (9.5) | |
| Established CHD | 15 (15.2) | 3 (4.8) | **0.04** |
| *CTCA findings* | *n = 84* | *n = 60* | |
| Total plaque volume, mm$^3$, median (IQR) | 29.5 (3.90–95.8) | 0.40 (0.0–7.4) | **< 0.001** |
| Calcified plaque volume, mm$^3$, median (IQR) | 20.8 (1.0–66.5) | 0.15 (0.0–7.1) | **< 0.001** |
| Soft/mixed plaque volume, mm$^3$, median (IQR) | 0.0 (0.0–8.68) | 0.0 (0.0–0.0) | **0.001** |
| CAC, Agatston units, median, (IQR) | 124 (8–534) | 1 (0–39) | **< 0.001** |
| *MetSO, AGEs (pmol/mg) and RAGE* | | | |
| MetSO | 61 ± 8 | 58 ± 9 | **0.020** |
| Glucosepane | 6480 ± 1254 | 3409 ± 607 | **< 0.001** |
| Pentosidine | 30 ± 10 | 18 ± 6 | **< 0.001** |
| Glucuronidine/LW-1 | 1045 ± 645 | 461 ± 319 | **< 0.001** |
| MG-H1 | 445 ± 193 | 299 ± 137 | **< 0.001** |
| Skin autofluorescence *arbitrary units* | 2.8 ± 0.53 | 2.2 ± 0.48 | **< 0.001** |
| sRAGE *pg/ml* | 1752 ± 973 | 1553 ± 541 | 0.14 |
| esRAGE *ng/ml* | 0.34 ± 0.29 | 0.28 ± 0.11 | **0.05** |

Data are mean ± SD or n (%) unless otherwise stated. There were 99 participants in the diabetes group and 63 participants in the control group, except for the CTCA findings as only the participants without established CHD were referred to CTCA. There were no missing data. CAD, coronary artery disease; IQR, inter-quartile range.
[a]estimated glomerular filtration rate calculated by MDRD formula

local anesthesia) and were excluded from the analyses leaving a total of 99 participants with diabetes with complete data (4% missing data). Fifteen participants in the diabetes group had established CHD, and the remaining 84 completed the CTCA.

As previously described, the diabetes and control groups were of similar age and sex distribution, and the diabetes group had lower LDL-cholesterol levels, higher systolic blood pressure

and lower diastolic blood pressure than the control group [4]. Five percent in the diabetes group had no retinopathy. The diabetes group had a significantly lower rate of normal coronary arteries than the control group (14.1% versus 47.6%) and a significantly higher rate of established CHD (15.2% versus 4.8%) (Table 1). CTCA also showed significantly higher rates of all plaque volume measures in the diabetes group compared to the control group (p≤ 0.001). All skin AGEs, MetSO, skin autofluorescence and the esRAGE were significantly higher in the diabetes group than in the control group (p ≤ 0.05 to < 0.0001) (Table 1 and S1 and S2 Figs).

## 3.2 Collagen MetSO, AGEs and serum RAGEs

In the diabetes group, all skin AGEs correlated with one another (r = 0.43–0.66, p < 0.001). MetSO correlated with MG-H1 (r = 0.42 p < 0.001) and glucosepane (r = 0.25, p = 0.02). Unlike MetSO and sRAGE/esRAGE, all four AGEs correlated with age (r = 0.25–0.48, p < 0.02) (Table 2). sRAGE and esRAGE correlated with eGFR (r = -.18, p = 0.02 and r = -0.26, p = 0.001 respectively). The other products did not correlate with eGFR, and there were no significant correlations with sex or disease duration. Among all markers, only skin glucosepane correlated significantly with mean $HbA_{1c}$ (r = 0.32, p = 0.003). In the control group, there was a significant correlation between glucosepane and pentosidine with age (r = 0.36 and 0.44, respectively, both p < 0.01), but not with $HbA_{1c}$-levels.

## 3.3 Associations of collagen markers with CAD

In the diabetes group, the participants with normal coronary arteries had lower levels of MetSO compared to the participants with any degree of CAD (mean ± SD 55.7 ± 9.6 versus 62.4 ± 7.5, respectively, p = 0.003) (Figs 1 and 2). This association remained significant when adjusting for age, sex and mean $HbA_{1c}$ (OR per 1SD increase in MetSO (0.48 (95% CI 0.27– 0.88), p = 0.02). None of the AGEs were significantly associated with having normal coronary arteries.

In the diabetes group, glucuronidine/LW-1 and skin autofluorescence were significantly higher in the group with established CHD compared to the participants without CHD entering the study. The mean ± SD glucuronidine/LW-1 levels in the patients with established CHD

**Table 2. Coefficients of correlation (Spearman's rho) in the diabetes group.**

|  | M/SPV | CPV | CAC | age | Sex | Mean HbA$_{1c}$ | MetSO | MG-H1 | glpane | pentos | LW-1 | sRAGE | esRAGE |
|---|---|---|---|---|---|---|---|---|---|---|---|---|---|
| TPV | .342** | .893** | .901** | .368** | .326** | .172 | .191 | .243* | .185 | .148 | .268* | .035 | .025 |
| M/SPV | - | .002 | .170 | .106 | .275* | .241* | .047 | .126 | .123 | .291** | .214 | -.067 | -.110 |
| CPV | - | - | .918** | .486** | .223* | .060 | .169 | .276* | .179 | .131 | .225* | .054 | .056 |
| CAC |  | - | - | .443** | .246* | .076 | .233* | .307** | .159 | .145 | .256* | .154 | .133 |
| Age | - | - | - | - | -0.026 | -.112 | .182 | .480** | .249* | .362** | .253* | .122 | .196 |
| Mean HbA$_{1c}$ | - | - | - | - | - | - | .144 | .120 | .318* | .055 | .087 | -.083 | -.121 |
| MetSO | - | - | - | - | - | - | - | .416** | .248* | .044 | .023 |  |  |
| MG-H1 | - | - | - | - | - | - | - | - | .661** | .615** | .434** |  |  |
| Glpane | - | - | - | - | - | - | - | - | - | .648** | .580** |  |  |
| Pentos | - | - | - | - | - | - | - | - | - | - | .593** |  |  |

* P < 0.05 (2-tailed).

**p < 0.01 (2-tailed).

TPV, Total plaque volume; M/SPV, Mixed/soft plaque volume; CPV, Calcified plaque volume; CAC, Coronary artery calcification score. MetSO, methionine sulfoxide; MG-H1, methylglyoxal hydroimidazolone; glpane, glucosepane; pentos, pentosidine; LW-1, glucuronidine/LW-1

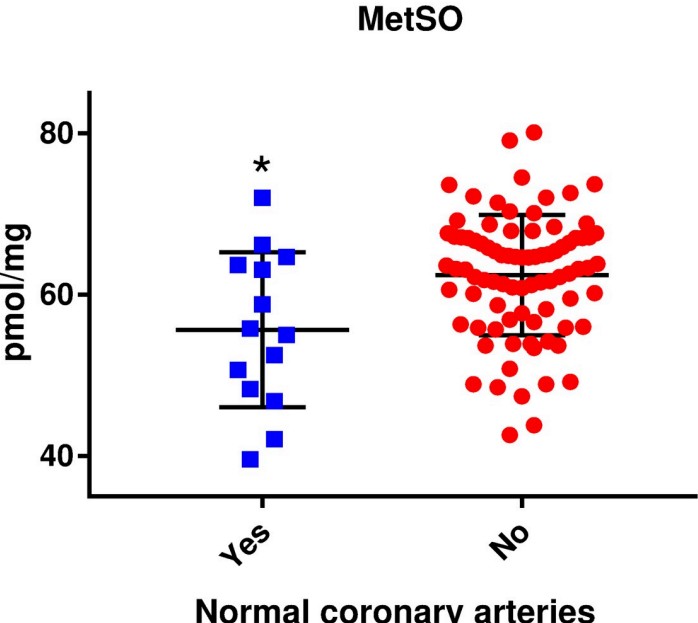

**Fig 1. Association between levels of MetSO and having normal coronary arteries.** Scatter dot plot with line at median with interquartile range. MetSO, methionine sulfoxide. *p = 0.02 compared to "No". Adjusted for age, sex and mean $HbA_{1c}$.

was 1495 pmol/mg ± 650 versus 965 pmol/mg ± 614, p = 0.003 in the participants without established CHD, and similarly for skin autofluorescence; 3.1 ± 0.66 versus 2.7 ± 0.48 (arbitrary units), p = 0.006. In the adjusted analyses (age, sex, mean $HbA_{1c}$) only glucuronidine/LW-1 (OR per 1SD increase in glucuronidine/LW-1 2.0 (95% CI 1.2–3.5), p = 0.01) was significantly associated with established CHD (Fig 2b). We also found that MG-H1 (p = 0.03) and MetSO (p = 0.01) were associated with retinopathy (background or proliferative) in the diabetes group in univariate analyses. In the adjusted analysis, the $OR_{adj}$ for retinopathy per 1SD increase in MetSO was 3.2 (95% CI 1.1–9.0). The association with MG-H1 did not reach statistical significance in the adjusted analysis (p = 0.06).

In the diabetes group, MG-H1 and glucuronidine/LW-1 correlated with both total plaque volume and calcified plaque volume (all p < 0.05) (Table 2/Fig 3b and 3c), however when controlling for age and sex, none of the associations remained statistically significant. Only pentosidine correlated with soft/mixed plaque volume, and this association remained significant after controlling for age, sex and mean $HbA_{1c}$ ($\beta_{ln}$ 0.06 (95% C.I. 0.02–0.09, p = 0.002). Glucosepane was not associated with any of the measures of CAD, and neither serum sRAGE nor esRAGE significantly correlated to any of the CAD measures (Table 2).

When performing the same analyses in the control group, we had the following significant findings: Pentosidine correlated with total plaque volume, also when adjusting for age and sex ($\beta_{ln}$ 0.09 (0.03–0.15, p = 0.007). Pentosidine also correlated with calcified plaque volume in the adjusted analysis ($\beta_{ln}$ 0.07 (0.01–0.13, p = 0.02). Skin autofluorescence correlated with total plaque volume only in the unadjusted analysis (r = 0.264, p = 0.04).

## 4. Discussion

In the present study cohort, CTCA allowed us to perform a detailed assessment of the coronary arteries, including plaque morphology. We demonstrated an association between low levels of

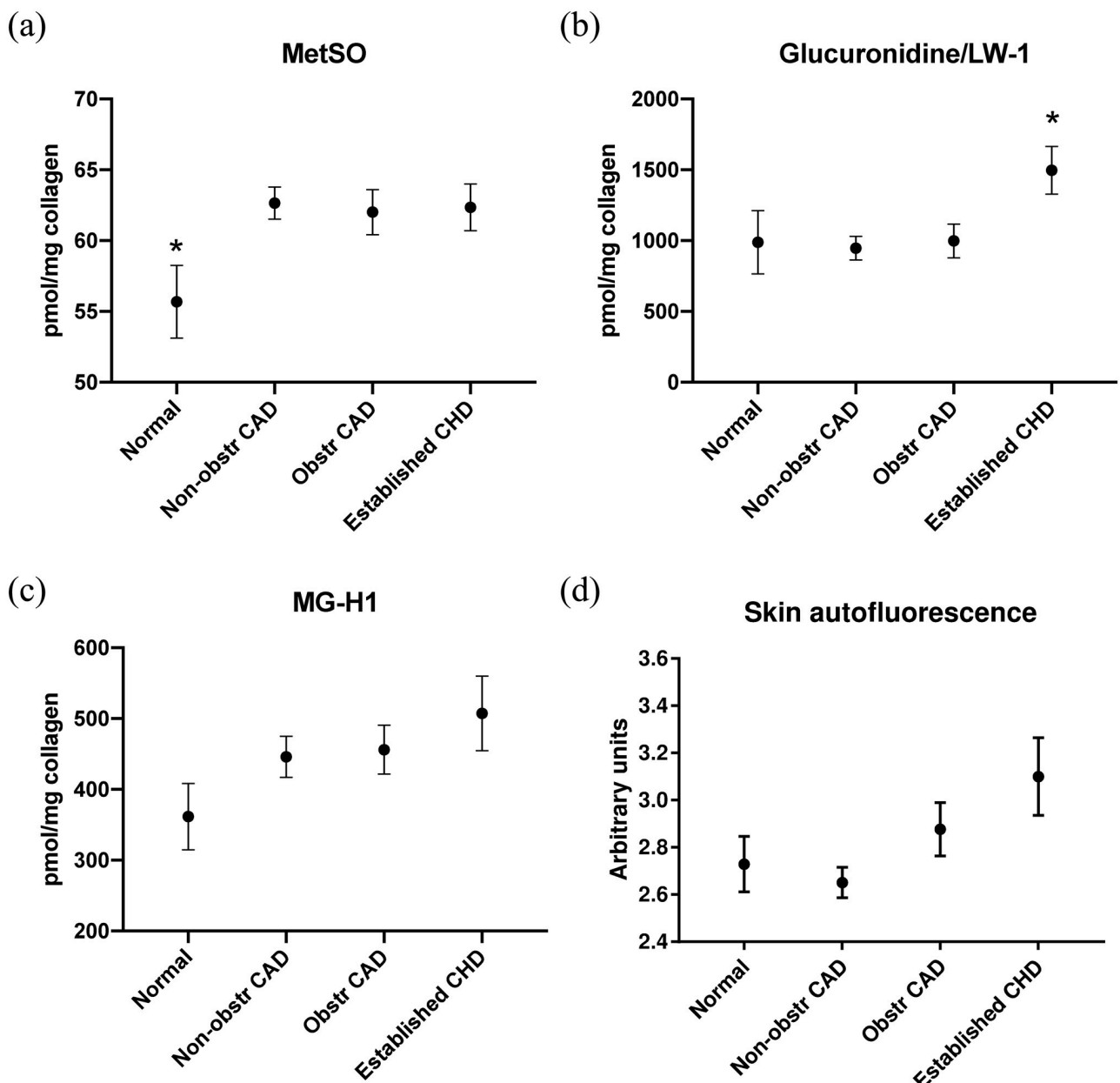

**Fig 2. Associations of MetSO, Glucuronidine /LW-1, MG-H1 and skin autofluorescence levels with categories of CAD in the diabetes group.**
Concentrations with standard errors of the mean of: 2a, MetSO; 2b, Glucuronidine /LW-1; 2c, MG-H1; and 2d, skin autofluorescence with CAD categorized as normal, non-obstructive CAD, obstructive CAD and established CHD in the diabetes group. MetSO, methionine sulfoxide; MG-H1, methylglyoxal hydroimidazolone; CAD, coronary artery disease. CHD, coronary heart disease. * P < 0.02 versus the other three categories after adjusting for age, sex and mean HbA$_{1c}$.

the oxidation product MetSO and having normal coronary arteries in patients with long-term type 1 diabetes. We also showed an association between glucuronidine/LW1 with established CHD and with total plaque volume, in particular calcified plaques; however the correlation in the latter analysis was not significant when adjusting for age and sex. Pentosidine was associated with soft/mixed plaque volume, also in the adjusted analyses. In the control group,

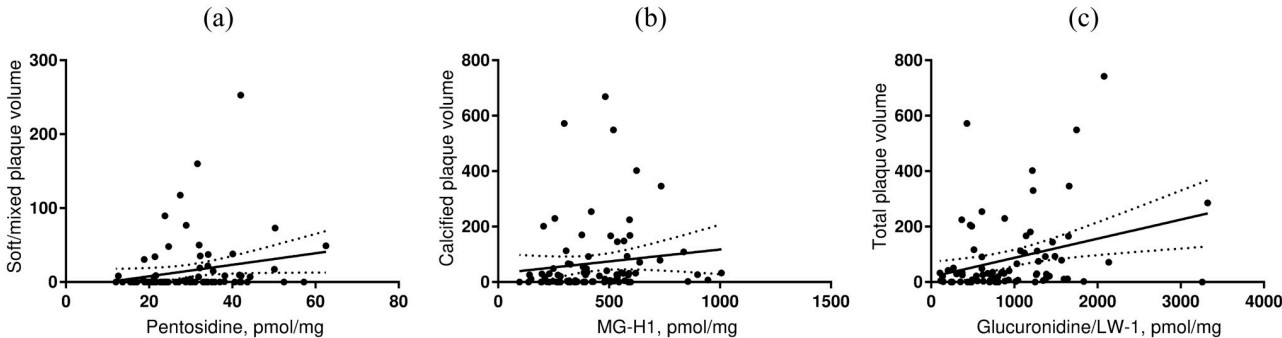

**Fig 3. Associations of pentosidine, MG-H1 and Glucuronidine /LW-1 with plaque volume in the diabetes group.** Scatterplots with linear regression lines and 95% confidence bands showing correlations between a, pentosidine and soft/mixed plaque volume; b, MG-H1 and calcified plaque volume; and c, glucuronidine/LW-1 and total plaque volume in the diabetes group. All correlations had a 2-tailed significance level at p < 0.05. The association between pentosidine and soft/mixed plaque volume was the only significant association after adjusting for age, sex and HbA$_{1c}$. MG-H1, methylglyoxal hydroimidazolone.

pentosidine was associated with total plaque volume, but there were no associations between MetSO and markers of CAD.

As expected, some patients with long-term type 1 diabetes remain free of CAD [4]. In the present study, we demonstrate that these patients have less accumulation of the collagen bound oxidation product MetSO in skin. This finding suggests that either oxidative collagen damage is associated with coronary atherosclerosis in type 1 diabetes or that MetSO is a marker of increased oxidative stress in the vascular bed. Protein methionine oxidation seems to have a role in vascular disease, and oxidation-sensitive methionine residues have been found in several proteins that are important in vascular biology, such as Apolipoprotein A-1, thrombomodulin and von Willebrand factor with resultant impaired or hyperactive protein function. [9] Interestingly, we found that low levels of collagen MetSO were not associated with having normal coronary arteries in the control group. However, we cannot exclude a type II error.

It is tempting to attribute the association between MetSO and CAD only in the participants with diabetes to chronic long-term hyperglycemia. Hyperglycemia increases the production of ROS in cultured bovine aortic endothelial cells, and the importance of oxidative stress in the formation of coronary atherosclerosis in diabetes has previously been identified in mechanistic studies in rodents [7, 21]. However, even though the half-life of skin collagen is about 15 years in individuals without diabetes [22], collagen bound MetSO did not correlate with mean 30-year HbA$_{1c}$ in our study. A reduced antioxidant defense in diabetes is supported by the significant association with pentosidine, the oxidation product of ascorbic acid. Alternatively, the impaired ability to respond to oxidative stress might explain why some patients with diabetes are more prone to developing coronary atherosclerosis [23].

No previous studies have reported on the association between collagen MetSO and detailed measures of coronary atherosclerosis assessed by CTCA. However, a study of 96 patients with type 1 diabetes showed an association between collagen MetSO and being prone to microvascular and macrovascular complications combined [10]. Also, the Oslo study demonstrated an association between collagen MetSO and arterial stiffness [11]. Conversely, a recent study showed a negative association between circulating MetSO levels and the incidence of cardiovascular events in type 2 diabetes [24]. Thus, there seemingly is paradoxical evidence in that circulating MetSO is negatively associated with cardiovascular disease while collagen

accumulated MetSO is positively associated with cardiovascular disease. Data suggest that methionine-residues on the surface of proteins can act as endogenous antioxidants through red-ox reactions (methionine sulfoxide reductase), whereas methionine-residues that are buried in the proteins are less prone to oxidation and more prone to being inactivated following oxidation. As such, oxidation of collagen methionine may be more likely to result in functional alterations of the protein, and collagen methionine sulfoxide could thus be seen as a marker of cumulative oxidative tissue damage [25]. In that regard, the diabetes participants who were free of retinopathy also had lower collagen MetSO levels underlining the importance of this marker in long-term complications.

A number of studies examining the association between AGEs and vascular complications do not specify the different AGEs, but look upon them as a whole. Rather, AGEs are a heterogeneous group, and the individual products most likely exhibit somewhat different pathogenic effects. Also, several of the studies linking AGE products to cardiovascular disease are on circulating AGEs [26–28]. Regarding collagen AGEs, Monnier et al. identified collagen glucuronidine/LW-1, pentosidine and MG-H1 to be the most important AGE markers for future subclinical cardiovascular disease, measured as carotid intima media thickness [12]. Previous studies have demonstrated that AGEs increase with age and duration [29], and indeed, the levels of AGEs in our study were higher than in previous studies with younger participants, exemplified by glucosepane (the most abundant AGE) which had a mean level of 6480 pmol ± 1254 in the diabetes group in our study compared to up to 4000 pmol/mg in patients with diabetes at age 30–40 years [17]. In the present study, we found that pentosidine and mean $HbA_{1c}$ correlated with soft/mixed plaques while MG-H1 and glucuronidine/LW-1 correlated with calcified plaques (the latter two in unadjusted analyses). Calcifications are common in progressive atherosclerotic lesions and the coronary artery calcification burden is greater in type 1 diabetes than in healthy controls [3, 30]. Eighty-three percent of plaques measured in our population with diabetes were calcified [15]. Although not as prone to rupture as soft plaques, calcified plaques/CAC are clinically important due to an association with a higher risk of future coronary events [31].

Glucuronidine/LW-1 was associated with both established CHD and calcified plaques in our study signifying a role in advanced stages of CAD. Recently, partial structure elucidation of glucuronidine/LW-1 revealed that it contains a glucuronide, suggesting a glucuronidine/LW-1 precursor is glucuronidated in the liver or kidney to form a circulating reactive glucuronide precursor that glycates collagen [32]. While the structure of the precursor is yet unknown, our findings strengthen previous reports of research that glucuronidine/LW-1 is associated with microvascular and macrovascular complications in type 1 diabetes [32, 33]. However, more research is needed into the process of glucuronidation and how this may be linked to complications in diabetes. Conceivably, excess glucuronidation of vital neutriceuticals could weaken antioxidant defenses and account for faster vascular complication progression in selected individuals [32, 34].

Skin autofluorescence has previously been suggested to be associated with both subclinical and clinical atherosclerosis independent of diabetes [35]. While skin autofluorescence was associated with total plaque volume only in unadjusted analyses in the control group, it was associated with established CHD in the diabetes group, again signifying a role in advanced stages of the disease.

In the present study, pentosidine was associated with soft/mixed plaque content, but not with calcified plaques. As ascorbic acid is a precursor for pentosidine-formation, this could suggest that increased ascorbic acid oxidation correlates with higher volume of soft/mixed plaque [36, 37]. The soft (non-calcified) lipid-laden plaques are important as they are more vulnerable to rupture causing cardiac events and will also respond better to intensified medical

treatment [30, 38]. Higher levels of circulating pentosidine have previously been associated with both incident fatal and non-fatal cardiovascular events and CAC in type 1 diabetes [27, 28]. The latter was not found in the present study. Our results add to the evidence that pentosidine is an important marker of CAD in type 1 diabetes.

Overall, the only similar findings in the diabetes and control groups were that pentosidine correlated with the total plaque volume in the control group and with mixed/soft plaque volume in the diabetes group. Measures of serum AGE levels, mainly reflecting N-(Carboxymethyl)lysine and to a smaller degree pentosidine, have previously been linked to severity of CAD on coronary angiography in participants without diabetes. However, the association between collagen pentosidine with measures of coronary atherosclerosis in persons without diabetes has to our knowledge, not been previously reported [39]. HbA$_{1c}$ did not correlate with pentosidine or with measures of coronary atherosclerosis in the control group. This indicates that the association between pentosidine and coronary plaques in the control group is at best indirectly related to hyperglycemia.

The present study was a cross-sectional study with its inherent limitations, and we can therefore not conclude that MetSO or the specific AGEs represent a cause or are a result of CAD. We did not find any significant associations between glucosepane and measures of CAD, on which the power analysis was based. Nevertheless, the power analysis was performed before data from the DCCT/EDIC study emerged, which identified other collagen AGEs (namely MG-H1, pentosidine and glucuronidine/LW-1) to be associated with measures of macrovascular disease [12]. Hence, while our findings must be interpreted as exploratory, they are in line with the data from the DCCT/EDIC study. The study is statistically relatively small and may be underpowered to detect weaker correlations. The control group were mostly spouses of the participants with diabetes, thereby having a similar lifestyle and diet, perhaps with similar intake of exogenous AGEs, which could possibly influence the total collagen AGE levels [40]. Still, all the AGEs were about twice the level in the diabetes group than the control group signifying the importance of hyperglycemia in endogenous AGE formation. Our biopsies were taken from skin collagen, and we do not know whether skin collagen is a good surrogate tissue for coronary arteries. However, based on previous observations that collagen glycation is more strongly correlated with long-term microvascular complications than hemoglobin glycation [12], accumulation of MetSO in long-lived skin collagen is expected to be a better reflection of arterial oxidative processes and damage than by similar processes in serum proteins. A further limitation is that we only have CTCA scans of the participants without established CHD who are selected from a survivor population. Hence, the correlations with plaque volume are therefore based on a population that probably has less CAD than the general type 1 diabetes population, possibly underestimating the effects of AGEs. Our participants with diabetes are from a single center, however as previously described, they had a similar HbA$_{1c}$ as the Norwegian population registered with the same duration [14]. Strengths of this study are a unique study population with very long-term type 1 diabetes with detailed measures of coronary atherosclerosis and chronic hyperglycaemia, a high inclusion rate of the eligible cohort and the presence of a control group.

## 5. Conclusions

In conclusion, we found an association between low levels of MetSO and normal coronary arteries on CTCA in long-term type 1 diabetes. There was also an association between high glucuronidine/LW-1 levels and advanced stages of CAD, i.e. established CHD and calcified plaques. Additionally, while MG-H1 and glucuronidine/LW-1 correlated with total and calcified plaque volume in univariate analyses, pentosidine correlated with soft/mixed plaques. Our

results strengthen the knowledge of the importance of metabolic and oxidative stress on vascular cells in the formation of atherosclerosis in diabetes. Future research on the metabolic and oxidative damage on coronary arteries may reveal pharmaceutical targets that can decrease the risk of CAD in type 1 diabetes.

## Supporting information

**S1 Fig. Scatter plots with linear regression lines showing the age-related accumulation of methionine sulfoxide (MetSO), glucuronidine/LW-1, methylglyoxal hydroimidazolone (MG-H1), pentosidine, glucosepane and skin autofluorescence in the type 1 diabetes group in red and controls in blue.**
(PDF)

**S2 Fig. Scatter plots with linear regression lines showing the age-related accumulation of sRAGE og esRAGE in the type 1 diabetes group in red and controls in blue.**
(PDF)

## Acknowledgments

The authors thank Anne Karin Molvær, research nurse and the other staff at NDC for administrative help. We thank Prof. Knut Dahl-Jørgensen for enthusiastic support during the years. We thank Morten Valberg and Cathrine Brunborg for help with statistical analyses (University of Oslo, Centre for biostatistics and epidemiology) and we thank all the participants in the study.

## Author Contributions

**Conceptualization:** Kristine B. Holte, Kristian F. Hanssen, Kari Anne Sveen, Tore Julsrud Berg.

**Data curation:** Kristine B. Holte, Mona Svanteson, Ingebjørg Seljeflot, David R. Sell, Vincent M. Monnier, Tore Julsrud Berg.

**Formal analysis:** Kristine B. Holte, Mona Svanteson, Tore Julsrud Berg.

**Funding acquisition:** Tore Julsrud Berg.

**Investigation:** Kristine B. Holte, Mona Svanteson, Ingebjørg Seljeflot, David R. Sell, Tore Julsrud Berg.

**Methodology:** Kristine B. Holte, Mona Svanteson, Kristian F. Hanssen, Ingebjørg Seljeflot, Svein Solheim, David R. Sell, Vincent M. Monnier, Tore Julsrud Berg.

**Project administration:** Kristine B. Holte, Tore Julsrud Berg.

**Resources:** Tore Julsrud Berg.

**Supervision:** Kristian F. Hanssen, Svein Solheim, Tore Julsrud Berg.

**Validation:** Kristine B. Holte, David R. Sell, Vincent M. Monnier, Tore Julsrud Berg.

**Writing – original draft:** Kristine B. Holte.

**Writing – review & editing:** Mona Svanteson, Kristian F. Hanssen, Kari Anne Sveen, Ingebjørg Seljeflot, Svein Solheim, David R. Sell, Vincent M. Monnier, Tore Julsrud Berg.

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
