## [Decision Letter · Decision Letter 0]

12 Feb 2020

PONE-D-20-00205

Collagen methionine sulfoxide and glucuronidine/LW-1 are markers of coronary artery disease in long-term survivors with type 1 diabetes. The Dialong study.

PLOS ONE

Dear Dr. Holte,

Thank you for submitting your manuscript to PLOS ONE. After careful consideration, we feel that it has merit but does not fully meet PLOS ONE’s publication criteria as it currently stands. Therefore, we invite you to submit a revised version of the manuscript that addresses the points raised during the review process.

We would appreciate receiving your revised manuscript by Mar 28 2020 11:59PM. To enhance the reproducibility of your results, we recommend that if applicable you deposit your laboratory protocols in protocols.io, where a protocol can be assigned its own identifier (DOI) such that it can be cited independently in the future. For instructions see: http://journals.plos.org/plosone/s/submission-guidelines#loc-laboratory-protocols

We look forward to receiving your revised manuscript.

Kind regards,

Ming-Chang Chiang

Academic Editor

PLOS ONE

Additional Editor Comments:

Dear Dr. Holte,

Thank you for submitting your manuscript to PLOS ONE. After careful consideration, we feel that it has merit but does not fully meet PLOS ONE’s publication criteria as it currently stands. Therefore, we invite you to submit a revised version of the manuscript that addresses the points raised during the review process.

Journal Requirements:

2. Thank you for including your ethics statement:  The regional ethics committee (project no. 2014/851) approved the study, and all participants signed a written informed consent form.

Reviewers' comments:

Reviewer's Responses to Questions

**Comments to the Author**

1. Is the manuscript technically sound, and do the data support the conclusions?

Reviewer #1: Yes

Reviewer #2: Partly

2. Has the statistical analysis been performed appropriately and rigorously? 

Reviewer #1: Yes

Reviewer #2: Yes

3. Have the authors made all data underlying the findings in their manuscript fully available?

Reviewer #1: Yes

Reviewer #2: Yes

4. Is the manuscript presented in an intelligible fashion and written in standard English?

Reviewer #1: Yes

Reviewer #2: Yes

5. Review Comments to the Author

Reviewer #1: In this cross-sectional study, the authors investigated the associations between the collagen oxidation products and advanced glycation end-products (AGEs) with measures of coronary artery diseases in individuals with type 1 diabetes with the duration of ≥ 45-years and controls. They found that Low levels of collagen-bound methionine sulfoxide were associated with normal coronary arteries while glucuronidine/LW-1 was positively associated with established CHD in long-term type 1 diabetes. The authors conclude that the dysregulation of metabolism and oxidative stress on vascular cells could be of importance in the formation of atherosclerosis in type 1 diabetes.

STRENGTHS:

(1) The unique study population with very long-term type 1 diabetes.

(2) Data on atherosclerosis in the long-standing T1D in populations are scarce.

COMMENTS AND QUESTIONS:

(1) Table 1. Have the authors measured other atherosclerosis risk factors, e.g. CRP? The subgroups of "Diabetes group" and "Controls without diabetes" may mislead the readers. I suggest the use of T1D instead of Diabetes. Are there eventual missing data? If yes, please specify it in the table note.

(2) Figure 2b. The axis title looks strange. I suggest inverting it.

Reviewer #2: The manuscript with a title of “Collagen methionine sulfoxide and glucuronidine/LW-1 are markers of coronary artery 2 disease in long-term survivors with type 1 diabetes. The Dialong study”. The authors have investigated the correlation between methionine sulfoxide and the collagen AGEs with coronary heart disease. There were more than 160 participants where HPLC and ELISA were used to measure the methionine sulfoxide and the collagen AGEs respectively. 99 participants of type 1 diabetes have coronary heart disease (CHD) or underwent Computed Tomography Coronary Angiography (CTCA). The novelty of this paper is mainly the correlation between MetSO and CAD and remaining correlations have been previously reported. L 427 says “Strengths of this study are a unique study population with very long-term type 1 diabetes with detailed measures of coronary atherosclerosis and chronic hyperglycaemia, a high inclusion rate of the eligible cohort and the presence of a control group”. Based on the that I believe this paper lacks novelty in the method and data section however the study population is unique although the small population used in this study! I recommend this work to be published in PLOSONE after addressing the below comments.

1. Add details for the MetSO, AGE’s and RAGE’s analysis method!

2. How the AGE’s in skin, RAGE’s in plasma are normalised?

3. L230: “all skin AGEs correlated with one another …” although this was found, only pentosidine correlation was found! Can author explain that!

4. L253: It seems that values of MetSO don’t match with table 1 - Can authors review explain?

5. L276: It seems that the glucuronidine/LW-1 and skin autofluorescence values don’t match with table 1 - Can authors review explain?

6. L349 “Thus, there seemingly is 349 paradoxical evidence in that circulating MetSO is negatively associated with cardiovascular 350 disease while collagen accumulated MetSO is positively associated with cardiovascular 351 disease” is it possible this applies for other AGE’s which will affect the correlation suggested by the authors?

7. L384 “In the present study, pentosidine was associated with soft/mixed plaque content, but 385 not with calcified plaques” the correlation between the plaque content with the pentosidine in skin are not direct because the circulating pentosidine was not measured? Can author elaborate in that?

8. Values in table 1, text and figures must be reviewed machining sure they all match.

9. L38 remove extra bracket.

6. PLOS authors have the option to publish the peer review history of their article (what does this mean?). If published, this will include your full peer review and any attached files.

Reviewer #1: No

Reviewer #2: No

---

## [Author Response · Author response to Decision Letter 0]

27 Mar 2020

We thank the academic editor and both the reviewers for thorough reviews of our manuscript and for the excellent comments. Please find the comments and our responses/action below.

Academic Editor’s comments and our responses:

We have reviewed PLOS ONE’s style requirements.

The name of the files have been changed/updated.

We have changed the wording of our ethics statement on page 5: “The Norwegian Regional Committees for Medical and Health Research Ethics - South East (project no. 2014/851) approved the study, and all participants signed an informed consent. “

3. PLOS only allows data to be available upon request if there are legal or ethical restrictions on sharing data publicly. In your revised cover letter, please address the following prompts:

We believe there are ethical and legal restrictions on sharing the dataset as outlined in the cover letter.

An ethically compliant dataset may be provided by the authors who can be contacted through dr Elisabeth Qvigstad at eqwigsta@ous-hf.no / elisabeth.qvigstad@medisin.uio.no.

Reviewer #1: In this cross-sectional study, the authors investigated the associations between the collagen oxidation products and advanced glycation end-products (AGEs) with measures of coronary artery diseases in individuals with type 1 diabetes with the duration of ≥ 45-years and controls. They found that Low levels of collagen-bound methionine sulfoxide were associated with normal coronary arteries while glucuronidine/LW-1 was positively associated with established CHD in long-term type 1 diabetes. The authors conclude that the dysregulation of metabolism and oxidative stress on vascular cells could be of importance in the formation of atherosclerosis in type 1 diabetes.  STRENGTHS: (1) The unique study population with very long-term type 1 diabetes. (2) Data on atherosclerosis in the long-standing T1D in populations are scarce.

Reviewer #1 comments and our responses:

(1) Table 1. 

-Have the authors measured other atherosclerosis risk factors, e.g. CRP? 

We have measured CRP which is also an interesting factor. It was higher in the T1D group (median 1.7 vs 1.2), but it did not correlate with any of our outcomes in the diabetes group.

We have added the median CRP levels to table 1.

-The subgroups of "Diabetes group" and "Controls without diabetes" may mislead the readers. I suggest the use of T1D instead of Diabetes. 

Thank you for pointing this out

We have changed the subgroups to “Type 1 diabetes” and “Controls”

-Are there eventual missing data? If yes, please specify it in the table note. 

Thank you for this important point. All participants were interviewed, examined, had skin biopsies taken, blood tests drawn and had either CTCA or had known CHD. There were no missing data.

We have added to the table note: “There were no missing data”

(2) Figure 2b. The axis title looks strange. I suggest inverting it.

We agree that the title of figure 2b should be inverted. 

The title now reads “Glucuronidine/LW-1”. We have also changed the label of the y-axis from pmol/mg to pmol/mg collagen” for Fig. 2a-2c. 

Reviewer #2: The manuscript with a title of “Collagen methionine sulfoxide and glucuronidine/LW-1 are markers of coronary artery 2 disease in long-term survivors with type 1 diabetes. The Dialong study”. The authors have investigated the correlation between methionine sulfoxide and the collagen AGEs with coronary heart disease. There were more than 160 participants where HPLC and ELISA were used to measure the methionine sulfoxide and the collagen AGEs respectively. 99 participants of type 1 diabetes have coronary heart disease (CHD) or underwent Computed Tomography Coronary Angiography (CTCA). The novelty of this paper is mainly the correlation between MetSO and CAD and remaining correlations have been previously reported. L 427 says “Strengths of this study are a unique study population with very long-term type 1 diabetes with detailed measures of coronary atherosclerosis and chronic hyperglycaemia, a high inclusion rate of the eligible cohort and the presence of a control group”. Based on the that I believe this paper lacks novelty in the method and data section however the study population is unique although the small population used in this study! I recommend this work to be published in PLOSONE after addressing the below comments.

Reviewer #2 comments and our responses:

1. Add details for the MetSO, AGE’s and RAGE’s analysis method!

Thank you for pointing this out. We agree that we can provide more details on the analysis method.

We have changed/added the following details to page 7:

Paragraph 2.4: “Skin collagen bound MetSO and AGEs (glucosepane, pentosidine, glucuronidine/fluorophore LW-1, and MG-H1) were measured using liquid chromatography tandem mass spectrometry (LC/MS/MS) as previously described (14). Briefly samples of 50 ug insoluble delipidated skin collagen samples were exhaustively digested with proteolytic enzymes to release the free AGEs. The samples were spiked with isotopically labeled internal standards for each AGE and quantitated by LC/MS/MS. Each chromatography peak was individually analyzed in reference to the labeled peak. Collagen content was determined using the hydroxyproline assay as previously described (17). Data are expressed in pmol AGE/mg collagen.”

Paragraph 2.5: “Serum sRAGE and esRAGE were quantified using enzyme-linked immunosorbent assay (ELISA) kits (Quantikine ELISA, R&D Systems, Abingdon, Oxon, UK and B-Bridge International Inc., Cupertino, CA, USA respectively) as directed by the manufacturer’s instructions”.

2. How the AGE’s in skin, RAGE’s in plasma are normalised?

Raw data are displayed in each figure. No normalization was performed, but data were statistically adjusted for various variables as described in the legends to the Tables and Figures.

3. L230: “all skin AGEs correlated with one another …” although this was found, only pentosidine correlation was found! Can author explain that!

Thank you for this interesting point. To further elaborate on this, we have added the full Spearman correlation coefficients for AGE to AGE comparison in the bottom part of Table 2.

AGEs are all raised in patients with diabetes, but they are a heterogenous group of compounds. For example, previous data suggest that glucosepane and MG-H1 are more strongly associated with microvascular disease and MG-H1 and LW-1 are more strongly associated with macrovascular disease (Ref 12). In our paper, LW-1/glucuronidine was associated with established CHD. Also, MG-H1 and LW-1/glucuronidine were associated with more advanced (calcified) plaques in unadjusted analyses. On the other hand, pentosidine was associated with soft/mixed plaque volume, also in the adjusted analysis. As further detailed in the paper, LW-1/glucuronidine seems to be a glucuronidation product whereas collagen pentosidine could reflect increased ascorbic acid oxidation. Hence, these AGEs could reflect different pathological processes in the formation of coronary plaques. 

See Table 2.

4. L253: It seems that values of MetSO don’t match with table 1 - Can authors review explain? 

Table 1 compares the value of MetSO in subjects with and without type 1 diabetes while L253 describes the level of MetSo only in participants with diabetes where the levels of MetSO in those with normal coronary arteries are compared to the levels of MetSO in those with any degree of CAD.

5. L276: It seems that the glucuronidine/LW-1 and skin autofluorescence values don’t match with table 1 - Can authors review explain?

Similarly, table 1 compares the value of glucuronidine/LW-1 in subjects with and without type 1 diabetes while L276 describes the level of glucuronidine/LW-1 only in participants with diabetes with established CHD versus those without established CHD.

6. L349 “Thus, there seemingly is 349 paradoxical evidence in that circulating MetSO is negatively associated with cardiovascular 350 disease while collagen accumulated MetSO is positively associated with cardiovascular 351 disease” is it possible this applies for other AGE’s which will affect the correlation suggested by the authors?

Thank you for this interesting comment. Collagen-linked MetSO represents a cumulative measure of oxidative damage to tissue over several years, while, as we suggest in the paper, methionine-residues on the surface of proteins (circulating MetSO) can act as endogenous antioxidants through red-ox reactions (ref 24). 

The relationship between collagen and circulating AGEs would be expected to be different. Further, the same inverse relationship has not been demonstrated with collagen and circulating AGEs. Indeed, while we found a positive correlation between collagen pentosidine and soft/mixed plaque volume, higher levels of circulating pentosidine have previously been associated with cardiovascular events (ref 26). 

It would be interesting to study the association between collagen and circulating AGEs, but this was beyond the scope of our study. 

7. L384 “In the present study, pentosidine was associated with soft/mixed plaque content, but 385 not with calcified plaques” the correlation between the plaque content with the pentosidine in skin are not direct because the circulating pentosidine was not measured? Can author elaborate in that? 

Thank you for this interesting point. We have not measured serum pentosidine, but previous studies have found an association between circulating pentosidine and both incident CVD events and CAC in type 1 diabetes (ref 26 and 27). 

It would have been interesting to have biopsies taken from the coronary arteries, and we do not know whether skin collagen AGE levels accurately represent collagen AGE levels in the vessel wall of the coronary arteries.

As skin collagen has a long half-life (ref 21), serum AGE levels and collagen AGE levels would be expected to differ and perhaps be different representations of pathological processes. Pentosidine is in all likelihood a marker of Vitamin C mishandling by the cell. Thus, elevated skin pentosidine may mean that chronic oxidative stress results in elevated levels of dehydroascorbate, i.e. the precursors of pentosidine. However, skin pentosidine can also form from Amadori product of glucose representing a cumulative form of glycemic insult. At this time, we do not have means to differentiate between ascorbic acid and glucose-derived pentosidine. 

8. Values in table 1, text and figures must be reviewed machining sure they all match.

We have reviewed all the values in the tables, figures and texts to make sure they all match

In paragraph 3.3, we changed the p-value from 0.017 to 0.02 to match the number of decimals given in figure 1 (p=0.02).

9. L38 remove extra bracket.

Thank you

We have removed the extra bracket.

---

## [Editor Report · Decision Letter 1]

3 Apr 2020

PONE-D-20-00205R1

Collagen methionine sulfoxide and glucuronidine/LW-1 are markers of coronary artery disease in long-term survivors with type 1 diabetes. The Dialong study.

PLOS ONE

Dear Dr. Holte,

Thank you for submitting your manuscript to PLOS ONE. After careful consideration, we feel that it has merit but does not fully meet PLOS ONE’s publication criteria as it currently stands. Therefore, we invite you to submit a revised version of the manuscript that addresses the points raised during the review process.

We would appreciate receiving your revised manuscript by May 18 2020 11:59PM. To enhance the reproducibility of your results, we recommend that if applicable you deposit your laboratory protocols in protocols.io, where a protocol can be assigned its own identifier (DOI) such that it can be cited independently in the future. For instructions see: http://journals.plos.org/plosone/s/submission-guidelines#loc-laboratory-protocols

We look forward to receiving your revised manuscript.

Kind regards,

Ming-Chang Chiang

Academic Editor

PLOS ONE

Additional Editor Comments (if provided):

In general, findings presented here are novel, experiments seem to be well done, and paper describes an interesting topic. However, there are some major points that complicate acceptance of the paper in the present format.

The editor makes the following suggestions:

1. Provide fluorescence images of Fig 2.

2. Provide protein levels of RAGE.

3. The discussion should be improved correlating these results with the literature.

4. The manuscript is not linked to current conversations in the journal.

---

## [Author Response · Author response to Decision Letter 1]

27 Apr 2020

We thank the editor for the thorough review of our manuscript and excellent comments. Please find our comments and changes to the manuscript below.

1. Provide fluorescence images of Fig 2.

Thank you for this suggestion. Figure 2 demonstrates the associations of a, MetSO, b, Glucuronidine/LW-1 c, MG-H1 and d, skin autofluorescence (measured using the AGE reader®) with categories of coronary artery disease in the diabetes group. MetSO, glucuronidine/LW-1 and MG-H1 were measured using LC-MS/MS. We agree that we could provide more raw data on the distribution of the levels of the variables, and we have now created scattergrams of the data versus age in both the diabetes and control groups as a supplement (Fig 1S).

 2. Provide protein levels of RAGE.

Thank you for this suggestion. We measured sRAGE and esRAGE in serum using ELISA. They are expressed in pg/ml (Table 1). We are also providing the scattergram of these data in Fig 2S.

 3. The discussion should be improved correlating these results with the literature.

We have added the following to the discussion: “Previous studies have demonstrated that AGEs increase with age and duration (29), and indeed, the levels of AGEs in our study were higher than in previous studies with younger participants, exemplified by glucosepane (the most abundant AGE) which had a mean level of 6480 pmol ± 1254 in the diabetes group in our study compared to up to 4000 pmol/mg in patients with diabetes at age 30–40 years (17).”

 4. The manuscript is not linked to current conversations in the journal.

We have added the following to the discussion which refers to a study by den Dekker MA et al, PLoS One 2013: “Skin autofluorescence have previously been suggested to be associated with both subclinical and clinical atherosclerosis independent of diabetes (35). While skin autofluorescence was associated with total plaque volume only in unadjusted analyses in the control group, it was associated with established CHD in the diabetes group, again signifying a role in advanced stages of the disease.”

---

## [Editor Report · Decision Letter 2]

30 Apr 2020

Collagen methionine sulfoxide and glucuronidine/LW-1 are markers of coronary artery disease in long-term survivors with type 1 diabetes. The Dialong study.

PONE-D-20-00205R2

Dear Dr. Holte,

We are pleased to inform you that your manuscript has been judged scientifically suitable for publication and will be formally accepted for publication once it complies with all outstanding technical requirements.

With kind regards,

Ming-Chang Chiang

Academic Editor

PLOS ONE

Additional Editor Comments (optional):

The authors complied with the experimental suggestions that now improve the experimental understanding and text. In the end I believe that the suggested modifications have been met making the work acceptable for publication.
---

## [Editor Report · Acceptance letter]

4 May 2020

PONE-D-20-00205R2 

Collagen methionine sulfoxide and glucuronidine/LW-1 are markers of coronary artery disease in long-term survivors with type 1 diabetes. The Dialong study. 

Dear Dr. Holte:

I am pleased to inform you that your manuscript has been deemed suitable for publication in PLOS ONE. Congratulations! Your manuscript is now with our production department. 

With kind regards,

on behalf of

Dr. Ming-Chang Chiang 

Academic Editor

PLOS ONE